# Abnormalities of IL-12 Family Cytokine Pathways in Autosomal Dominant Polycystic Kidney Disease Progression

**DOI:** 10.3390/medicina60121971

**Published:** 2024-11-30

**Authors:** Corina-Daniela Ene, Ilinca Nicolae, Cristina Căpușă

**Affiliations:** 1Department of Internal Medicine and Nephrology, Carol Davila University of Medicine and Pharmacy, 050474 Bucharest, Romania; ccalexandr@yahoo.com; 2Nephrology Department, Dr Carol Davila Clinical Hospital of Nephrology, 010731 Bucharest, Romania; 3Research Department, Victor Babes Clinical Hospital of Infectious Diseases, 030303 Bucharest, Romania; drnicolaei@yahoo.ro

**Keywords:** ADPKD, pathogenesis, IL-12, inflammation, renal cysts

## Abstract

*Background and Objectives:* Autosomal Dominant Polycystic Kidney Disease (ADPKD) is the most frequent genetic renal disease with a complex physiopathology. More and more studies sustain that inflammation plays a crucial role in ADPKD pathogenesis and progression. We evaluated IL-12 involvement in ADPKD pathophysiology by assessing the serum levels of its monomers and heterodimers. *Materials and Methods:* A prospective case-control study was developed and included 66 ADPKD subjects and a control group of 40 healthy subjects. The diagnosis of ADPKD was based on familial history clinical and imagistic exams. The study included subjects with eGFR > 60 mL/min/1.73 mp, with no history of hematuria or other renal disorders, with stable blood pressure in the last 6 months. We tested serum levels of monomers IL-12 p40 and IL-12 p35 and heterodimers IL-12 p70, IL-23, IL 35, assessed by ELISA method. *Results:* IL-12 family programming was abnormal in ADPKD patients. IL-12p70, IL-12p40, and IL-23 secretion increased, while IL-12p35 and IL-35 secretion decreased compared to control. IL-12p70, IL-12p40, and IL-23 had a progressive increase correlated with immune response amplification, a decrease of eGFR, an increase in TKV, and in albuminuria. On the other hand, IL-35 and IL-12p35 were correlated negatively with CRP and albuminuria and positively with eGFR in advanced ADPKD. *Conclusions:* The present study investigated IL-12 cytokine family members’ involvement in ADPKD pathogenesis, enriching our understanding of inflammation in the most common renal genetic disorder.

## 1. Introduction

Macrophages are a diverse group of cells within the mononuclear phagocytic system, contributing crucially to homeostasis, remodeling, and immune regulation, being involved in the pathogenesis of most renal diseases. Several kidney diseases require significant involvement of both M1 and M2 macrophages. Macrophages perform unique functions when exposed to different stimuli in the specific microenvironment of kidney diseases. M1 macrophages secrete pro-inflammatory cytokines (such as IL-6, IL-1, IL-23, IL-12, MMP12, iNOS, and MINCLE) and contribute to renal tissue damage, inflammation, and fibrosis, while M2 macrophages play an anti-inflammatory role and are involved in immunosuppression, matrix remodeling, tissue repair, and regeneration [1,2,3,4,5]. Renal tubular cells and immune cells seem to be interconnected, generally through growth factors and cytokines, to orchestrate signaling cascades in renal disease progression [6,7,8,9,10,11,12,13].

Nowadays, ADPKD pathogenesis is not fully understood. ADPKD is the most common genetic disorder, characterized by persistent involvement of the immune system, tubular epithelial cell proliferation, fibrosis, interstitial inflammation, focal renal hypoxia, extracellular matrix remodeling, normal renal parenchyma destruction with cysts formation, kidney volume increase and progression to end stages of chronic kidney disease [6,9,11,12,13,14,15,16,17,18]. Considering the inflammatory nature of ADPKD, the cytokine system could represent a target of current research focused on the pathogenesis of this disease. Tubular epithelial cells and mesangial cells are an intrinsic source of IL-12 in renal diseases [19,20,21]. It has been reported that IL-12 plays an important role in chronic proliferative renal lesions, tubular atrophy, glomerular or interstitial inflammation, and cystogenesis [18,20,21,22,23,24,25,26,27].

IL-12 is a family of proinflammatory cytokines produced by dendritic cells and activated phagocytes. Its family is formed from five natural heterodimeric members, biologically active, noted as: IL-12 (IL-12p35/IL-12p40), IL-23 (IL-23p19/IL-12p40), IL-27 (IL-27p28/Ebi3), IL-35 (IL-12p35/Ebi3), IL-39 (IL-19p19/Ebi3) and two synthetic members noted IL-X (Ebi3/IL-23p19) and IL-Y (IL-12p40/IL-27p28) [11]. Each member contains an alpha subunit (p19, p28, p35) and a beta subunit [p40, protein 3 induced by Epstein-Barr (EBI3)]. The biogenesis, quaternary structure, specific receptors, signaling pathways, and biological activities of IL-12 family members are intensively studied nowadays in inflammation, autoimmunity, and onco-immunology [28,29,30,31,32,33,34,35,36,37,38,39]. IL-12 signals are transmitted through IL-12Rβ1 and IL-12Rβ2; IL-23 signals through IL-12Rβ1 and IL-23R; IL-27 uses gp130 and IL-27R (WSX-1); while IL-35 uses gp130 and IL-12Rβ2 [21]. IL-12 activates T, NK, and CD8+ cells, promoting proliferation and cytotoxicity, inducing gamma interferon production (IFNγ), regulating the transition from innate to adaptative immunity, promoting T-cells mediated cytolysis, stimulating directly dendritic cells to produce IL-12 interleukin and to present the antigen [3,5,20].

New studies showed that IL-12 family members have different effects on the immune system; IL-12 promoted Th1 cell synthesis, while IL-23 was involved in Th17 cell stabilization, T memory cell activation, and neutrophils IL-17 mediated recruitment in the inflammation process. Although IL-12 and IL-23 induced different immune responses, they both could be characterized as proinflammatory. Meanwhile, IL-12 could play a critical role in regulating the equilibrium between pro- and anti-inflammatory signaling by inducing inhibition of Th1 and Th17 differentiation. Moreover, IL-35 seemed to function as an anti-inflammatory cytokine by stimulating T-regulating cells and IL-35 and IL-10. IL-39 had proinflammatory effects by neutrophil expansion and differentiation [3,4,5,20].

The interactions between different cells involved in ADPKD physiopathology are orchestrated by soluble mediators [1,2,3,4,5]. The present study assessed the serum levels of IL-12 family monomers (IL-12p35, IL-12p40) and heterodimers (IL-12p70, IL-23, IL-35) and their relation to the clinical and biological status of ADPKD subjects and debated IL-12 involvement in ADPKD physiologypathology.

## 2. Materials and Methods

Patient characteristics. We developed a prospective, case-control study in Carol Davila Clinical Hospital of Nephrology that included 117 subjects: a group of 72 ADPKD subjects (46.3 years old, men:women 35:32), and a group of 45 healthy subjects (41.4 years old, men:women 21:24), similar as sex, mean age. The patient characteristics are presented in Table 1.

ADPKD diagnosis was based on familial history, clinical exam, and CT or MRI scan. The total kidney volume was calculated after imagistic exams in every ADPKD subject. The values for kidney volume presented in the control group are based on data in the literature. We excluded from the study subjects that presented vascular aneurysms detected on CT/MRI, cysts in other organs except the kidney, with eGFR < 45 mL/min/1.73 mp, high levels of C reactive protein (CRP) with a history of hematuria, cysts infection, urinary tract infection, renal lithiasis, with unstable blood pressure in the last 6 months. Regarding the genetic testing, 52% were genetically tested for PKD1 and PKD2 genes (Table 1).

The subjects included in the study were stratified by eGFR according to the KDIGO CKD stages and CRP. CRP levels between 3 and 10 mg/L define low-grade inflammation, and between 10 and 50 mg/L define high-grade inflammation (HGI). CRP levels higher than 50 mg/dL are often associated with acute bacterial infections, and these subjects are excluded from the study. More and more studies have shown that both C3 and C4 serum levels increase in inflammation [39]. These molecules were assessed in ADPKD and control groups as markers of inflammation, but no statistical differences were detected.

Laboratory determinations. The serum levels of monomers IL-12 p40 and IL-12 p35 and heterodimers IL-12 p70, IL-23, IL 35 were assessed in the study. The blood samples were collected after subjects fasting during 12 h, using a holder–vacutainer system. The blood was kept for one hour at room temperature and then centrifugated, afterwards, the sera were separated and frozen at −80 degrees. Hemolyzed, icteric, lactescent, or microbiologically contaminated samples were excluded.

IL-12 monomers and heterodimers were assessed by the ELISA method using the following ELABSCIENCE kits: HumanIL-12p70 ELISA kit, Cat.No.:E-EL-H0150; HumanIL-12p35 ELISA kit, Cat.No.:E-EL-H1647, HumanIL-12p40 ELISA kit Cat.No.:E-EL-H0151, HumanIL-23 ELISA kit Cat.No.:E-EL-H0107 and HumanIL-35 ELISA kit Cat.No.:E-EL-H2443. A capture antibody is highly specific for each member of the IL-12 cytokine family that has been coated to the wells of the microtiter strip plate provided following incubation. The first amplification step is performed by adding the Biotine-Tyramine reagent. A biotin polymerization reaction occurs in the region of the HRP linked to the detection antibody. After washing, the second amplification step is performed, and the polymerized biotin is revealed by a new streptavidin-HRP step. Finally, after washing, a chromogen substrate is added to the wells, resulting in the progressive development of a colored complex with the conjugate, its development being stopped by the addition of acid. The intensity of the complex is directly proportional to the concentration of cytokine in the samples. The absorbance of the color complex is then measured using the TECAN analyzer.

Statistical analysis. All the results were analyzed using IBM SPSS Statistics 2015 and were presented as average value ± standard deviation. *p* < 0.05 was considered with statistical significance. Data were compared using ANOVA, with Tukey’s post hoc test or the Kruskal–Wallis test for normally distributed data, or, for non-normally distributed data, using Dunn’s post hoc test. The relation between the studied parameters was evaluated by Pearson’s correlation coefficient before The Kolmogorov–Smirnov test was used to evaluate the data normality.

## 3. Results

### 3.1. IL-12 Cytokine Status in ADPKD

In ADPKD patients, we detected the medium eGFR lower and TKV significantly higher than in the control group; complement (C3, C4) and albumin decreased, while CRP increased significantly when compared with the control group. Urinary b2-microglobulin and urinary albumin to creatinine ratio were statistically significantly higher in ADPKD when compared with the control group.

In ADPKD patients, we detected altered production of IL-12 cytokine (Table 2). Though IL-12p70 increased 2.12-folds (*p* < 0.05), IL-12p35 decreased 1.29-folds (*p* > 0.05), IL-12p40 was 3.88-folds increased (*p* < 0.05), IL-23 increased 2.56-folds (*p* < 0.05), and IL-35 decreased 1.38-folds (*p* > 0.05) in ADPKD compared to control group.

### 3.2. IL-12 Related Cytokine According to eGFR in ADPKD Patient

We analyzed the serum levels of IL-12 family members in relation to eGFR. The ADPKD subjects were grouped according to eGFR: G1—eGFR > 90 mL/min/1.73 m^2^, G2—eGFR between 60–89 mL/min/1.73 m^2^ and G3—eGFR between 45–59 mL/min/1.73 mp. IL-23 levels were 1.63 folds lower in the G3 group compared to G1; IL-12p35 levels were 1.58 folds lower in the G3 group compared to G1, while IL-35 levels were 1.71 folds lower in the G3 group compared to G1. IL-12p70 and IL-12p40 levels decreased while eGFR increased (Table 3). IL-12p70 was 1.43 folds higher in G3 compared to G1, and IL-12p40 was 4.67 folds higher in G3 compared to G1.

### 3.3. IL-12 Related Cytokine According to CRP Levels in ADPKD Patients

When analyzing IL-12 family members in relation to CRP, we detected the highest levels of IL-12p70, IL-12p40, and IL-23 and the lowest levels of IL-12p35 and IL-35 in the high inflammation group (Table 4). IL-12p70 increased by 1.17 folds in MI and 1.26 folds in HI compared with NI groups. IL-12p35 decreased by 1.14 folds in MI and 1.87 folds in HI compared with NI groups. IL-12p40 increased 3.95 folds in MI and 4.67 folds in HI compared with NI groups. IL-23 increased by 2.41 folds in MI and 1.97 folds in HI compared with NI groups. IL-35 decreased by 2.47 folds in MI and 1.08 folds in HI compared with NI groups.

### 3.4. IL-12 Cytokine Family Members in Relation with Clinical and Paraclinical Features of ADPKD

Regarding the relation between IL-12 family members, IL-12 p70 correlated negatively with IL-12p35 while IL-35 correlated positively with IL-12p40 and IL-23 (Table 5). IL-12p35 correlated positively with IL-12p40 and IL-35 and negatively with IL-23, while IL-23 correlated negatively with IL-35 (Table 5). IL-12p40 correlated positively with IL-23 and negatively with IL-35 but without statistical significance (Table 5).

We analyzed the relation between IL-12 cytokines and clinical and paraclinical features of ADPKD, as presented in Table 6. IL-12p70 correlated positively and statistically significantly with CRP, TKV, and UACR and negatively with C3, C4, and eGFR. IL-12p35 correlated negatively with CRP, ESR, C3, and TKV and positively with C4, urinary beta2-microglobuline, UACR, and eGFR. IL-12p40 correlated positively, statistically significant with CRP, TKV, and negatively with eGFR. IL-23 correlated positively with CRP, ESR, TKV, and UACR and negatively with eGFR, while IL-35 correlated negatively with CRP, TKV and positively with UACR and eGFR.

## 4. Discussion

ADPKD is the most frequent genetic renal disease with a complex physiopathology. Mutations in one of PKD1 or PKD2 genes that encode the proteins polycystin-1 and polycystin-2 account for most cases of ADPKD. These mutations induce abnormal cell functions and cyst formation in many organs, cysts that grow progressively and affect the structure and function of these organs. In kidneys, the cysts induce stress on surrounding tissue, modify signaling pathways, and induce fibrosis [3,4,5,11,34,35,36,37,38,39]. Moreover, cilia dysfunction, defective removal of cyst proteins in the kidneys, PC1 and PC2 influence on cell signaling pathways, key transcription factors combined with epithelial dedifferentiation, nephrotoxicity, ischemia, compensatory hypertropia, hyperglycemia could play an important role in initiating and development of cysts in ADPKD [5,39,40]. Molecular mechanisms like proliferation, inflammation, cell differentiation, cytokines, and growth factors could permit tissue remodeling and repairment of renal lesions [5,40,41].

The present study showed an alteration of IL-12 cytokine family members’ status in ADPKD subjects (Table 7). In ADPKD subjects with eGFR > 90 mL/min/1.73 sqm, IL-12 cytokines had higher serum levels compared to healthy subjects. In subjects with lower eGFR, progressive increase of IL-12p70, IL-12p40, and IL-23 secretion and decrease of IL-12p35 and IL-35 secretion compared to control were associated with ADPKD progression. The molecular events that induce overexpression or suppression of IL-12 cytokine family production in ADPKD are not fully understood. IL-12 cytokines could be secreted both after immune and non-immune stimulation [20,35,42,43]. It should be noted that IL-12, IL-23, and IL-27 are primarily derived from immune cells (effector T lymphocytes, macrophages, and dendritic cells), while IL-35 is produced by helper T cells [42,43,44].

IL-12 family members had different expressions in ADPKD subjects, and they were linked with this disorder progression. The present study aimed to decipher the nature, roles, and involvement mechanisms of each member of the IL-12 family in the regulation of cyst formation and renal function decline in ADPKD. Serum variations of IL-12 cytokines were strongly associated with kidney volume, especially when eGFR was low. An increase in kidney volume is a marker of the progression of disease in ADPKD. The strong positive correlation between IL-12p70, IL-23, and TKV suggests the involvement of these molecules in disease progression. Meanwhile, the relation between IL-35 and TKV was statistically negative, denoting a role in limiting cyst evolution (Table 7). Based on the biological effect of each immunocytokine, IL-12 family members could be considered to have double roles in cyst formation and kidney injury. IL-23 mediates cyst formation via IL-12/IFN, IL-23/IL-17 STAT/JAK signaling [36,43], while IL-35 suppresses Th cell activity [3].

Macrophages play a key role in renal fibrosis. Macrophage-related cytokines facilitate cystic renal cell proliferation or renal fibrosis in ADPKD via TGF beta, IL-17 TNF alpha, or IFNγ [2,20,45]. IL-12 elicits kidney injury by fostering the accumulation of IFN-gamma [35,46,47,48,49]. IL-23 acts directly on podocytes via Rac1, actin, and synaptopodin [50]. IL-23 mediates cyst formation. via IL-12/IFN, IL-23/IL-17,STAT/JAK signaling [34,41]. The results of the present study suggest the involvement of IL-12 family members in mechanisms of renal cyst development. Many molecules affect TKV increase, like growth factors (epidermic growth factors, insulin-like growth factor) and activators of adenylyl-cyclase (arginine, vasopressin, secretin, adenosine, and adrenaline [51]. Renal cysts secrete cytokines and chemokines while they grow, inducing inflammation and fibrosis in surrounding interstitial tissue, blocking urine flow in renal tubes, and, finally, determining apoptosis of upstream tubular segments. ADPKD progression consists of renal cysts and kidney volume increase while eGFR decreases [52]. TKV, a marker of disease progression, was negatively associated with filtration rate, albuminuria, interstitial inflammation, fibrosis, and glomerulosclerosis [51].

**Table 7 medicina-60-01971-t007:** Multifaceted IL-12family- cytokines in ADPKD progression.

Kidney Status	A Portrait of the IL-12 Family in ADPKD	Results of Present Research	Potential Mechanisms
ADPKD pathogenesis	The serum pattern of IL-12 family was modified in ADPKD subjects versus control. Accentuation or suppression of IL-12 members secretion could be associated with ADPKD progression.	IL-12p70, IL-12p40 and IL-23 were increased in contrast to low IL-35 and IL-12p35 ADPKD compared to control (Table 2, Table 3 and Table 4).	Immune cells infiltration and renal cells produce IL-12-cytokine family [34,53,54].
Renal cystogenesis	IL-12p70, IL-12p40, IL-23 initiate renal cysts formation. IL-12p35, IL-35 attenuate cysts growth and disease progression.	IL-12p70, IL-12p40, IL-23 were overexpressed (Table 2, Table 3 and Table 4), while IL-12p35, IL-35 were supressed in ADPKD (Table 3 and Table 4). Serum levels of IL-12 family were associated with TKV (Table 6).	Macrophages-related cytokines facilitate cystic renal cell proliferation or renal fibrosis in ADPKD via TGF beta,. IL-17 TNF alpha, IFNγ [2,45] IL-23 mediate cyst formation. via IL-12/IFN, IL-23/IL-17,STAT/JAK signaling [33,41]. IL-35 supress Th cells activity) [44].
Kidney function impaired	IL-12 cytokine family induce renal injury. Renal function decrease while cysts grow	UACR was positively correlated with IL-23 (Table 6). eGFR and TKV were significantly correlated with (Table 6). IL-12p40 varied iscordantly with eGFR (Table 5).	IL-12 elicits kidney injury by fostering the accumulation of IFN-gamma [45,46,47,48,49,53,54]. IL-23 acts directly on podocytes, via Rac1, actin and synaptopodin [54].
Inflammatory profile	IL-12p70, IL-12p40, IL-23 act as independent active molecule in relation to IL-12p35, IL-35 in ADPKD	IL-12p70, IL-12p40, IL-23h have divergent activities in relation to IL-12p35, IL-35 (Table 5): positive correlation between IL-12p70, IL-23, IL-12p40 and CRP (Table 6), and negative between IL-35, IL-12p35 and CRP in advanced APKDS (Table 6).	IL-12 family orchestrate arginine metabolism via NOS or ARG [33]. IL-12 stimulates Th1. IL-23 modulates CaMK4, ARG1, Th17, RORγT [55]. IL-12p40 and IL-12p80 are antagonists of IL-12Rbeta1 [33,50,52,56,57,58]. IL-12p35, IL-I3, IL-35 are anti-inflammatory molecules that supress IL-12 bioactivity by inhibiting the association between IL-12 and IL-12R β2 [41,44].
Outstanding question.	IL-12family cytokines has a causal role in ADPKD susceptibility?	Future results.	Future studies shall aim to understand the relationship and hierarchy between genetic data, defect in biogenesis and dynamics of IL-12 in ADPKD states [41,59].

The regulatory effect of IL-12 cytokines on inflammatory response in ADPKD was strongly investigated in the present study. IL-12p70, IL-12p40, and IL-23 had a progressive increase correlated with immune response amplification, a decrease of eGFR, an increase in TKV, and in albuminuria. On the other hand, IL-35 and IL-12p35 were correlated negatively with CRP and albuminuria and positively with eGFR in advanced ADPKD. Accordingly, IL-12p70, IL-12p40, and IL-23 had divergent activities in relation to IL-12p35 and IL-35 in modulating the immune response in ADPKD. Our results are sustained by the data published in other recent studies. Renal cysts secrete cytokines and chemokines during their increase, inducing fibrosis and inflammation in surrounding interstitial tissue. IL-12 and IL-23 are considered to play a pro-inflammatory role that amplifies the tissue lesions mediated by immune responses. IL-35 acts as an anti-inflammatory molecule that protects against inflammatory signals, while IL-27 has a double role in modulating inflammation, acting as an anti-inflammatory/pro-inflammatory molecule, depending on the inflammatory environment [51,60]. IL-12p70 activates NK cells, inducing IFN-gamma and promoting renal fibrosis. IL-12p40 is capable of upregulating pro-inflammatory TNF-alfa, inducing interstitial inflammation and fibrosis. IL-23 regulates T memory cells, activates macrophages, and maintains chronic autoimmune inflammation. Based on our results, IL-35 could be evaluated in further studies as a potential therapeutic target for modulating the inflammatory response in ADPKD.

IL-12 cytokine family orchestrates arginine metabolism via NOS or ARG [33,34]. IL-12 stimulate Th1; IL-23 modulateCaMK4, ARG1, Th17, RORγT [25]. IL-12p40 and IL-12p80 are antagonists of IL-12Rbeta1 [34,50,51]; IL-12p35, IL-23, IL-35 are anti-inflammatory molecules that suppress IL-12 bioactivity via inhibition the association between IL-12 and IL-12R β2 [43,60,61]. Other studies confirm the role of cytokines in renal inflammation. ADPKD subjects had significantly higher levels of IL-6, IL-8, MCP-1, TNF-α, and IFN-γ compared with healthy subjects [62,63]. Based on these findings, our study includes IL-12 cytokines family in the class of renal inflammation regulators, besides other molecules, such as pentraxins, ILs (1-β, -4, -8, -9, -10, -17f, -18), Interferon γ, IL-1 receptor antagonist, TGF-β, promarker D panel (ApoA4, CD5L, C1QB, IBP-3) adipokines and related compounds, visfatin, resistin, leptin, vascular cell adhesion molecule-1, E-selectin, neopterin (monocyte/macrophage activator) [64,65].

More and more data suggest that inflammation plays a crucial role in ADPKD progression. The role of interstitial inflammation in cyst development and ADPKD evolution to end-stage renal disease was studied in human and animal models. Proinflammatory cytokines IL-1β, TNF-α, and IL-2 were identified in the cyst fluid of human APKD kidneys [63,65]. Inflammation in ADPKD involves a complex interaction between different types of cells (B and T cells, macrophages, dendritic cells, interstitial fibroblasts, tubular epithelial cells), cytokines, chemokines, complement, and different signaling pathways [55,63,66,67,68,69]. All these factors could trigger and perpetuate inflammation, inducing interstitial lesions, apoptosis, and progressive interstitial fibrosis [55,66,70]. High mitochondrial stress via the NLRP3 inflammasome pathway activates the production of proinflammatory cytokines and oxygen-reactive species. Moreover, dysregulation of lipidic metabolism in immune cells triggers an inflammatory response in the kidneys [59,71,72,73,74].

The present findings support the notion that inflammation is an important element of ADPKD physiopathology, being the first study in the literature that evaluates the role of IL-12 cytokines family members in ADPKD, a very complicated disease characterized by complex interactions between genetics and variables that cannot be satisfyingly described by quantitative relations. We described at one point the serum levels of IL-12 cytokine family members, their relationship with clinical and paraclinical parameters (eGFR, TKV), and inflammatory markers using only simple regression. We consider this fact a limitation of the study, and a multivariable regression analysis that includes several potential markers should be presented in a further study. Altered serum levels of IL-12 cytokine members in ADPKD would be interesting to be correlated with these molecule urinary levels. While the study evaluated a limited number of urinary markers, future studies to evaluate the role of urinary cytokines levels will be developed. The urinary levels of IL-12 family members could correlate positively with their serum levels in ADPKD and could predict disease progression.

The study followed a control group with healthy subjects and an ADPKD group with a limited number of subjects, an aspect that limits the strength of the study. A clear contrast to the ADPKD group would be provided when comparing with control groups of subjects with other cystic or tubule-interstitial diseases. Though a more comprehensive multivariate approach, the inclusion of control groups with other cystic or tubule-interstitial diseases could provide a clearer understanding of the inflammatory dynamics in ADPK.

## 5. Conclusions

The present study investigated IL-12 cytokine family member involvement in ADPKD pathogenesis, enriching our understanding of inflammation in the most common renal genetic disorder. ADPKD is a cellular source of IL-12 cytokines produced by antigenic stimulation. IL-12 family programming was abnormal in ADPKD patients, where we detected synergic and divergent action of IL-12 cytokine family members, aspects that sustain the crucial role of inflammation in ADPKD physiopathology.

## Figures and Tables

**Table 1 medicina-60-01971-t001:** Participants status.

Characteristics	ADPKD	Control	*p* Value
Age(years)	47.3 (36.8–52.5)	41.4 (33–48)	NS
M/F	35/37	21/24	NS
eGFR (mL/min/1.73 m^2^)	78 (54.2–102.8)	102 (89–114)	<0.01
TKV (mL)	1659 (304–1856)	119 (100–200) ^x^	<0.001
Genetic analysis (PKD1/PK2)	38 (31/7)	-	NS
Hypertensive patients (%)	3 (4.5%)	1 (2.5%)	NS
Diabetic patients (%)	6 (8.3%)	2 (5%)	NS
Cardiovascular diseases (%)	7 (9.7%)	2 (5%)	NS
Patients with haematuria (%)	27 (43.7%)	1 (2.5%)	<0.01
Albumin (g/dL)	3.7 (2.9–4.5)	4.4 (3.6–5.33)	<0.01
CRP (mg/dL)	3.4 (0.5–14)	0.1 (0.0–0.8)	0.001
ESR (mm/h)	18 (2–41)	9 (1–12)	0.07
Urinary b2-microglobulin (mg/L)	3.3 (1.0–4.40)	0.04 (0.03–0.12)	<0.001
UACR (mg/g creatinine)	15 (4–23)	6 (5.5–7)	<0.001
C3 (mg/dL)	47 (28–105)	92 (70–120)	<0.01
C4 (mg/dL)	21 (9–38)	47 (38–60)	<0.01

ADPKD—Autosomal Dominant Polycystic Kidney Disease, M/F—Male/Female, TKV—total kidney volume, CRP—C Reactive Protein, ESR—Erythrocytes Sedimentation Rate; C3 and C4—Complement; UACR—Urine Albumin-Creatinine Ratio, eGFR—estimated Glomerular Filtration Rate, ^x^—medium value according to data in literature.

**Table 2 medicina-60-01971-t002:** IL-12A and IL-12B-derived cytokine levels in ADPKD patients and control.

Parameters	ADPKD Group	Contol Group	*p* Value
IL-12p70 (pg/mL)	38.0 ± 14.1	17.9 ± 4.7	0.027
IL-12p35 (pg/mL)	20.3 ± 3.8	26.2 ± 10.7	0.113
IL-12p40 (pg/mL)	299.3 ± 89	77.3 ± 19.2	0.001
IL-23 (pg/mL)	79.4 ± 27.1	31.0 ± 6.8	0.007
IL-35 (pg/mL)	8.4 ± 0.8	11.6 ± 6.5	0.381

ADPKD—Autosomal Dominant Polycystic Kidney Disease; IL—Interleukin; p40 and p35 subunits of IL-12 family; *p*—statistical significance.

**Table 3 medicina-60-01971-t003:** IL-12A and IL-12B related cytokine values according to eGFR in ADPKD patients.

Parameters	eGFR (mL/min/1.73 m^2^)			*p* Value	
	45–59 (G3)	60–89 (G2)	>90 (G1)	P_1_	P_2_
Patients number	19	35	22		
IL-12p70 (pg/mL)	52.9 ± 8.1	42.3 ± 10.2	36.8 ± 12.1	0.04	G1vsG2 = 0.24 G1vsG3 = 0.007 G2vsG3 = 0.002
IL-12p35 (pg/mL)	18.8 ± 4.4	26.6 ± 9.3	29.8 ± 8.1	0.01	G1vsG2 = 0.009 G1vsG3 = 0.0003 G2vsG3 = 0.419
IL-12p40 (pg/mL)	435.2 ± 102.1	368.0 ± 87.5	93.0 ± 7.1	0.04	G1vsG2 = 0.028 G1vsG3 = 0.0006 G2vsG3 = 0.0006
IL-23 (pg/mL)	59.1 ± 16.2	78.2 ± 31.4	96.9 ± 35.4	0.04	G1vsG2 = 0.007 G1vsG3 = 0.037 G2vsG3 = 0.0002
IL-35 (pg/mL)	7.8 ± 3.1	11.9 ±4.5	13.4 ± 5.7	0.01	G1vsG2 = 0.036
					G1vsG3 = 0.047 G2vsG3 = 0.229

ADPKD—Autosomal Dominant Polycystic Kidney Disease; IL—Interleukin; p70 and p40—subunits of IL-12; eGFR—estimated Glomerular Filtration Rate; G1—eGFR > 90 mL/min/1.73 m^2^, G2—eGFR between 60–89 mL/min/1.73 m^2^ and G3—eGFR between 45–59 mL/min/1.73 mp. according to KDIGO guidelines; *p*—significance level, P1—triple/comparison of the groups, P2—pairwise comparison of the groups.

**Table 4 medicina-60-01971-t004:** IL-12A and IL-12B related cytokine values according to CRP levels in ADPKD patients.

Parameters	CRP (mg/L)			*p* Value	
	>10 (HI)	3–10 (MI)	<3 (NI)	P_1_	P_2_
Patients number	19	31	16		
IL-12p70 (pg/mL)	46.9 ± 13.1	43.8 ± 18.6	37.2 ± 17.4	0.04	G1vsG2 = 0.24 G1vsG3 = 0.008 G2vsG3 = 0.004
IL-12p35(pg/mL)	17.2 ± 3.9	28.1 ± 3.11	32.2 ± 5.9	0.01	G1vsG2 = 0.001 G1vsG3 = 0.0008 G2vsG3 = 0.319
IL-12p40(pg/mL)	435.2 ± 102.1	368.0 ± 87.5	93.0 ± 7.1	0.04	G1vsG2 = 0.028 G1vsG3 = 0.0003 G2vsG3 = 0.0007
IL-23(pg/mL)	91.1 ± 21.6	74.4 ± 13.7	37.7 ± 9.1	0.011	G1vsG2 = 0.001 G1vsG3 = 0.0009 G2vsG3 = 0.002
IL-35 (pg/mL)	5.9 ± 1.8	13.5 ± 4.7	14.6 ± 4.8	0.033	G1vsG2 = 0.0004
					G1vsG3 = 0.0006 G2vsG3 = 0.219

ADPKD—Autosomal Dominant Polycystic Kidney Disease; IL-Interleukin; CRP—C-reactive protein; (*p*—significance level, P1—triple/comparison of the groups, P2—pairwise comparison of the groups.

**Table 5 medicina-60-01971-t005:** Connection between IL-12 cytokine family members in patients with ADPKD.

Parameters		IL-12p70	IL-12p35	IL-12p40	IL-23
IL-12p35	R	−0.22	-	-	-
	P	0.034	-	-	-
IL-12p40	R	0.044	0.067	-	-
	P	0.322	0.565	-	-
IL-23	R	0.512	−0.362	0.077	-
	P	0.0004	0.010	0.436	-
IL-35	R	−0.411	0.675	−0.065	−0.443
	P	0.036	0.002	0.718	0.006

ADPKD—Autosomal Dominant Polycystic Kidney Disease; IL—Interleukin; p70 and p40—subunits of IL-12; R—coefficient correlation; P—level of statistical significance

**Table 6 medicina-60-01971-t006:** Connection between IL12A and IL-12B –derived cytokines and clinical course in patients with ADPKD.

		IL-12p70	IL-12p35	IL-12p40	IL-23	IL-35
CRP	R	0.44	−0.23	0.52	0.63	−0.62
	P	0.002	0.05	0.03	<0.01	<0.01
ESR	R	0.21	−0.28	0.22	0.18	−0.46
	P	0.15	0.34	0.83	0.08	0.34
C3	R	−0.51	−0.16	−0.12	0.23	0.07
	P	0.04	0.74	0.72	0.43	0.95
C4	R	−0.28	0.27	0.19	0.21	0.10
	P	0.06	0.88	0.81	0,85	0.98
TKV	P	0.16	−0.45	0.47	0.58	−0.33
	R	0.02	0.064	0.050	<0.01	0.041
Urinary b2-microglobuline	P	−0.33	0.32	0.09	0.22	0.18
	R	0.22	0.37	0.95	0.31	0.56
UACR	R	0.41	0.26	−0.22	0.44	0.07
	P	0.06	0.11	0.17	0.046	0.93
eGFR	R	−0.48	0.44	−0.43	−0.51	0.27
	P	0.022	0.09	0.02	0.017	0.028

ADPKD—Autosomal Dominant Polycystic Kidney Disease; IL—Interleukin; p70 and p40—IL-12 family; CRP—C Reactive Protein; ESR—Erythrocyte Sedimentation Rate; C3, C4—Complement; TKV—total kidney volume; UACR—Urine Albumin-Creatinine Ratio; eGFR—estimated Glomerular Filtration Rate; R—coefficient correlation; P—level of statistical significance.

## Data Availability

All the data are presented in the paper.

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
