# Peer review of "Abnormalities of IL-12 Family Cytokine Pathways in Autosomal Dominant Polycystic Kidney Disease Progression"

_medicina, 2024, doi:10.3390/medicina60121971_

Round 1

Reviewer 1 Report

Comments and Suggestions for Authors

Review for Medicina on the article:

“Abnormalities of IL-12 Family Cytokine Pathways in Autosomal Dominant Polycystic Kidney Disease Progression”

Corina-Daniela Ene 1,2*, Ilinca Nicolae 3 and Cristina Capusa 1,2

The authors have conducted an extensive discussion on the role of interleukins and inflammations in ADPKD patients. The theme is very interesting and should be consider for several future studies. The role of IL-12 cytokines family is still under debate in this disease.

There are many limitations on the study, quite explained by the authors.

I have a few concerns on the study:

- did the patients have vascular aneurysms? Considering IL12 mostly secreted by macrophage, even vascular abnormalities could have an impact on those levels

- the limited number of individuals make the analysis quite incomplete.

- it would be very interesting to have even a genetic data on these patients. This feature could add a lot to the study.

- another important feature that could enrich the work should be to determine the same interleukins on urine samples

- considering all the limitations, the goal of the study should be to debate on the subject and not to drive conclusions on the matter.

Author Response

Dear Reviewer,

Thank you for giving me the opportunity to submit a revised draft of my manuscript titled Abnormalities of IL-12 Family Cytokine Pathways in Autosomal Dominant Polycystic Kidney Disease Progression to Medicina, Urology and Nephrology Section. We appreciate the time and effort that you have dedicated to providing your valuable feedback on the manuscript. We are grateful for the insightful comments on the paper. We have been able to incorporate changes to reflect most of the suggestions. We have highlighted the changes within the manuscript.

Here is a point-by-point response to your comments and concerns.

Comment 1:

- did the patients have vascular aneurysms? Considering IL12 mostly secreted by macrophage, even vascular abnormalities could have an impact on those levels

We added in the exclusion criteria the presence of vascular aneurysms.

Comment 2:

- the limited number of individuals make the analysis quite incomplete.

We mentioned these aspects as a limitation of the study.

Comment 3:

- it would be very interesting to have even a genetic data on these patients. This feature could add a lot to the study.

We have added some data about genetic tests in ADPKD subjects in Table 1. Genetic testing is still not affordable in our country, though only 52% of ADPKD subjects had genetic tests.

Comment 3

- another important feature that could enrich the work should be to determine the same interleukins on urine samples

We have mentioned this aspect as a study limitation. We explained the potential role of urinary levels of IL-12 family member in ADPKD progression. We will continue our research  on urinary interleukins in ADPKD.

Comment 4

- considering all the limitations, the goal of the study should be to debate on the subject and not to drive conclusions on the matter.

We reformulated the goal of the study.

Reviewer 2 Report

Comments and Suggestions for Authors

Dear authors,

This study sheds light on the role of inflammation in the progression of ADPKD (autosomal dominant polycystic kidney disease) and the important role played by the IL-12 cytokine family in its pathophysiology. However, it leaves several profound questions and areas for further consideration, suggesting potential directions for future research. However, there are some areas for improvement and comments.

1. The study presents an in-depth analysis of serum IL-12 family cytokines (IL-12p70, IL-12p40, IL-12p35, IL-23, IL-35) and their differential expression in ADPKD patients compared to healthy controls. The fact that IL-12p70, IL-12p40, and IL-23 are upregulated, while IL-12p35 and IL-35 are downregulated in ADPKD patients, is consistent with the hypothesis that inflammation plays a significant role in ADPKD pathogenesis. However, the potential dual roles of these cytokines, acting both pro-inflammatory and anti-inflammatory, need further clarification.

2.  While the study provides a thorough exploration of IL-12 cytokine family members, the authors acknowledge several limitations, such as the use of only simple regression analysis and the limited number of urinary markers. Additionally, the control group was limited to healthy subjects, which may not fully capture the inflammatory profile of other kidney diseases that could resemble ADPKD.

3.  Given that IL-35 was found to be negatively correlated with inflammatory markers (CRP) and positively correlated with eGFR in advanced ADPKD, can IL-35 be considered a potential therapeutic target for modulating the inflammatory response in ADPKD? Would strategies to enhance IL-35 expression or activity potentially slow disease progression?

4.  The study highlights the need to investigate urinary cytokines as potential biomarkers of disease progression. How do serum and urinary levels of IL-12 family members correlate, and could urinary cytokine profiles provide a more accurate reflection of local inflammation and kidney damage?

5.  The study discusses how IL-12 cytokines may promote fibrosis through macrophage activation and other inflammatory processes. What are the specific pathways by which IL-12p70, IL-12p40, and IL-23 contribute to renal fibrosis, and could blocking these cytokines prevent or reduce fibrosis in ADPKD?

6.  The study acknowledges the limitation of using only a healthy control group and simple regression models. Could a more comprehensive multivariate approach, or inclusion of control groups with other cystic or tubule-interstitial diseases, provide a clearer understanding of the inflammatory dynamics in ADPKD?

Author Response

Dear Reviewer,

Thank you for giving me the opportunity to submit a revised draft of my manuscript titled Abnormalities of IL-12 Family Cytokine Pathways in Autosomal Dominant Polycystic Kidney Disease Progression to Medicina, Urology and Nephrology Section. We appreciate the time and effort that you have dedicated to providing your valuable feedback on the manuscript. We are grateful for the insightful comments on the paper. We have been able to incorporate changes to reflect most of the suggestions. We have highlighted the changes within the manuscript.

Here is a point-by-point response to your comments and concerns.

  1. The study presents an in-depth analysis of serum IL-12 family cytokines (IL-12p70, IL-12p40, IL-12p35, IL-23, IL-35) and their differential expression in ADPKD patients compared to healthy controls. The fact that IL-12p70, IL-12p40, and IL-23 are upregulated, while IL-12p35 and IL-35 are downregulated in ADPKD patients, is consistent with the hypothesis that inflammation plays a significant role in ADPKD pathogenesis. However, the potential dual roles of these cytokines, acting both pro-inflammatory and anti-inflammatory, need further clarification.

We explained the dual role of these cytokines in Discussion section.

2.  While the study provides a thorough exploration of IL-12 cytokine family members, the authors acknowledge several limitations, such as the use of only simple regression analysis and the limited number of urinary markers. Additionally, the control group was limited to healthy subjects, which may not fully capture the inflammatory profile of other kidney diseases that could resemble ADPKD.

We have mentioned these when we presented the limitations of the study.

3.  Given that IL-35 was found to be negatively correlated with inflammatory markers (CRP) and positively correlated with eGFR in advanced ADPKD, can IL-35 be considered a potential therapeutic target for modulating the inflammatory response in ADPKD? Would strategies to enhance IL-35 expression or activity potentially slow disease progression?

We consider that IL-35 could be considered a potential therapeutic target as mentioned in Discussion section.

4.  The study highlights the need to investigate urinary cytokines as potential biomarkers of disease progression. How do serum and urinary levels of IL-12 family members correlate, and could urinary cytokine profiles provide a more accurate reflection of local inflammation and kidney damage?

The urinary levels of IL-12 family members would be increased and could correlate positively with serum levels of these markers in ADPKD subjects. High urinary levels of these molecule could predict ADPKD progression.

5.  The study discusses how IL-12 cytokines may promote fibrosis through macrophage activation and other inflammatory processes. What are the specific pathways by which IL-12p70, IL-12p40, and IL-23 contribute to renal fibrosis, and could blocking these cytokines prevent or reduce fibrosis in ADPKD?

L-12p70 activates NK cells, inducing IFN-gamma and promoting renal fibrosis. IL-12p40 is capable to upregulate pro-inflammatory TNF-alfa inducing interstitial inflammation and fibrosis. IL-23 regulates T memory cells, activate macrophages and maintain chronic autoimmune inflammation.

6.  The study acknowledges the limitation of using only a healthy control group and simple regression models. Could a more comprehensive multivariate approach, or inclusion of control groups with other cystic or tubule-interstitial diseases, provide a clearer understanding of the inflammatory dynamics in ADPKD?

We consider that a more comprehensive multivariate approach, or inclusion of control groups with other cystic or tubule-interstitial diseases could provide a clearer understanding of the inflammatory dynamics in ADPKD. We have mentioned this in discussions.

Reviewer 3 Report

Comments and Suggestions for Authors

Authors of this prospective, case control study evaluated IL-12 involvement in ADPKD pathophysiology by assessing the serum levels of its monomers and heterodimers. For this purpose, 66 ADPKD subjects and a control group of 40 healthy subjects were included in the study. They discovered, among others, that IL-12 family programming was abnormal in ADPKD patients: IL-12p70, IL-12p40 and IL-23 secretion increased, while IL-12p35 and IL-35 19 secretion decreased compared to control. 

It is an interesting study, in general, investigating potential pathopysiological mechanisms in this most common renal genetic disorder that has profound clinical consequences regarding outcome, need for treatment of associated sequelae (such as hypertension, renal function deterioration etc.) as well as regarding halthcare burden the disease presents to a society. It provides a novel insight in some aspects of this disease that tries to enrich our understandings about inflammation in this disease.

However, there are some issues that must be addressed by the authors:

1. Since this is a genetic disorder, its etiology is known. Therefore, study of cytokins profile in this disease may enrich our knowledge about it to some extent, however, it gives no practical possibillity to influence treatment options. Authors should, therefore, better explain the purpose of the study, in addition to gain knowledge about disease pathophysiology.

2. Materials and Methods

In Table 1, values of C3 and C4 are presented. Authors should explain in more detail the rationale why they included this in analysis. Namely, both C3 and C4 values are within normal limits in ADPKD patients (average C3 value slightly decreased) as well as in control subjects.

In addition, some patients characteristics should be added, such as medications they use or comorbidities as these may potentially influence results.

3. Results 

3.1. IL-12 cytokine status in ADPKD, 

1st paragraph: 

In ADPKD patients we detected the medium eGFR, TKV significantly higher than in control group; ...........

- Comment: it should be written: "In ADPKD patients we detected the medium eGFR lower and TKV significantly higher than in control group ..."

4. There are several grammatical errors, including some  words in Romanian language in otherwise Englist text. In addition, some results, presented in tables as well as  some parts of discussion text are very cumbersome, complicated and difficult to understand and should be shortened and rewritten.

Author Response

Dear Reviewer,

Thank you for giving me the opportunity to submit a revised draft of my manuscript titled Abnormalities of IL-12 Family Cytokine Pathways in Autosomal Dominant Polycystic Kidney Disease Progression to Medicina, Urology and Nephrology Section. We appreciate the time and effort that you have dedicated to providing your valuable feedback on the manuscript. We are grateful for the insightful comments on the paper. We have been able to incorporate changes to reflect most of the suggestions. We have highlighted the changes within the manuscript.

Here is a point-by-point response to your comments and concerns.

  1. Since this is a genetic disorder, its etiology is known. Therefore, study of cytokins profile in this disease may enrich our knowledge about it to some extent, however, it gives no practical possibillity to influence treatment options. Authors should, therefore, better explain the purpose of the study, in addition to gain knowledge about disease pathophysiology.

 We explained the aim of the study and the impact of study results on ADPKD physiopathology and dynamics.

  1. Materials and Methods

In Table 1, values of C3 and C4 are presented. Authors should explain in more detail the rationale why they included this in analysis. Namely, both C3 and C4 values are within normal limits in ADPKD patients (average C3 value slightly decreased) as well as in control subjects.

We explained why C3 and C4 values were presented In Material and Method Section.

In addition, some patients characteristics should be added, such as medications they use or comorbidities as these may potentially influence results.

 We added comorbidities in Table 1.

  1. Results 

3.1. IL-12 cytokine status in ADPKD, 

1st paragraph: 

In ADPKD patients we detected the medium eGFR, TKV significantly higher than in control group; ...........

- Comment: it should be written: "In ADPKD patients we detected the medium eGFR lower and TKV significantly higher than in control group ..."

We corrected

  1. There are several grammatical errors, including some  words in Romanian language in otherwise Englist text. In addition, some results, presented in tables as well as  some parts of discussion text are very cumbersome, complicated and difficult to understand and should be shortened and rewritten.

We corrected the grammatical errors and the words in Romanian language.

We shortened and rewritten some parts of Results and Discussion, as underlined in the manuscript.